# Computation of the Electrical Resistance of a Low Current Multi-Spot Contact

**DOI:** 10.3390/ma15062056

**Published:** 2022-03-10

**Authors:** Gideon Gwanzuwang Dankat, Laurentiu Marius Dumitran

**Affiliations:** Laboratory of Electrical Materials, Faculty of Electrical Engineering, University Politehnica of Bucharest, Splaiul Independentei 313, 060042 Bucharest, Romania; gdankat@elmat.pub.ro

**Keywords:** electrical contact, contact resistance, electrical connector, numerical analysis, FEM, COMSOL Multiphysics

## Abstract

In high complexity electrical systems such as those used in the automotive industries, electric connectors play an important role. The automotive industry is gradually shifting its attention to electric cars, which means more electrical connectors for sensors and data collection. A fault in connectors for sensors used in a vehicle can cause drastic damage to capital equipment and, in the worst case, the loss of life. The studies of faults or degradation of electrical contacts are essential for safety in vehicles and various industries. Although such faults can be due to numerous factors (such as dust, humidity, mechanical vibration, etc.) and some yet to be discovered, high contact resistance is the main factor causing erratic behavior of electrical contacts. This paper presents a study on the computation of electrical contact resistance of two metal conductors (in the form of a disk) with analytical relations and a numerical computation model based on the finite element method (FEM) in COMSOL Multiphysics. The contact spots were considered to have a higher electrical resistivity value (ρ_cs_) than those of the two metal conductors (ρ_Cu_). Studies such as the one in view that is carried out on a microscopic level are often difficult to investigate experimentally. Therefore, with the help of a simplified numerical model, the consequences of the degradation of electrical contacts are investigated. To validate the FEM model, the numerical results were compared to those obtained from analytical models.

## 1. Introduction

Electric contacts are one of the essential components in electronics and automotive systems. Estimating the contact resistance determines the accuracy of connectors. In machinery assembly, evaluating the reliability and tightness of contact can be done using the contact resistance of metals contact surface. For better conductivity and lower resistance, the point of contact between two bodies must have a large area with fewer impurities [1].

The growing number of electronic devices that equip electric vehicles calls for an increase in the number of electric connectors to be adapted in the vehicles’ electrical system. The automotive market demands optimum safety requirements and software-enabled features at a low price; to attain this, automotive manufacturers are trying to miniaturize electrical contacts to reduce weight by manufacturing them with high-performing alloys such as CuNiSi (copper: balance; 0.8–1.8% nickel; 0.01–0.05% phosphorus; and 0.15–0.35% silicon). Additionally, aluminum or copper alloys (CuMg, CuAg) are used for smaller wire cross-sections to reduce weight [2]. An important aspect of electrical contact technology is contact housing. They are usually made of polymer and perform functions such as electrical shielding, insulation, and protection of all connector components. Contact between two conductors occurs at discrete spots owing to the rough nature of the surfaces (asperities). Therefore, the real point of contact is a smaller portion of the apparent contact area [3].

Holm [4] in the 1930s presented a well-known theory describing contact systems and contact resistance. Based on several hypotheses, Holm’s model was established for the contact resistance of a single contact spot without considering the effect of surface films. Greenwood [5] presented a more elaborate interpretation of Holm’s model; he derived a formula for calculating contact resistance based on the number of a set of circular contact spots and the distance between them. Similar to Holm’s model, the influence of interface films was ignored.

Over time, Holm and Greenwood’s theory has been further developed and serves as a starting point for researchers in designing quality and reliable electrical contacts. Boyer [6] generalized the Greenwood formula in his analytical solution; he introduced the influence of interface films. Nakamura and Minowa [7,8,9,10] computed numerically the contact resistance using the boundary element method (BEM) and finite element method (FEM) of a contact model consisting of two cubic electrodes that communicate through square spots [8,9]. Nakamura also analyzed the contact resistance of regular and irregular conducting spots and their behavior [10]. Timsit [11] examined the electrical conduction through small constrictions and the dependence of electrical resistance on the shape and dimensions of the contact spots [12]. Shujuan et al. [13] built a contact resistance model based on a rough surface contact model by surface profile measurement and statistical analysis. Ren et al. [14] and Lim et al. [15] both analyzed contact resistance by applying a combination of experimental, numerical, and analytical methods. Malucci [16,17] established several models, considering the influence of interface films to predict the performance of degraded contacts. Furthermore, Fukuyama et al. [18] evaluated the contact resistance of a simplified contact sample and compared the result to the contact resistance calculated using Holm and Greenwood equations.

Although few studies of contact resistance focus on comparing numerical simulation and analytical solution, this paper presents a simplified contact model to evaluate the contact resistance by analytical approaches (using Holm equation and Greenwood equation) and by numerical simulation (using FEM in COMSOL Multiphysics). The model consists of two metallic disks coming in contact through multiple regular circular spots (a-spots) that have a higher electrical resistivity compared to the two metallic disks (ρ_cs_ < ρ_Cu_) in view to simulate the contact aging. Though analytical models, in theory, are considered to be the exact solution, they are derived based on multiple assumptions. Despite all the assumptions considered in the analytical models, the present study aims to explore the limits of numerical calculation for the contact resistance in the case of a simplified model. Moreover, this paper examines the influence of the size of the contact spots on the value of contact resistance.

### Aging of Electrical Contacts

Electrical contacts undergo different stresses during operation (i.e., mechanical wear due to induced vibrations, atmospheric conditions, corrosion, etc.). Swingler et al. [19] classified these stresses on automotive connectors into two significant groups. One is external stresses due to the environmental conditions, while the other is internal stresses created by the vehicle. Studies on contact resistance calculations and degradation of electrical contacts can be very tasking and complex because of the contact interface’s coarse nature and asperities that comprise numerous spots constricting the flow of electrons and the numerous degrading factors manifesting at the interface.

Environmental factors such as temperature, air pollution, humidity, condensation, etc., affect electrical connectors because they promote the rate of corrosion (Figure 1); Corrosion is much more severe at the interface of connectors; it gradually reduces the area of contact at the interface, which increases resistance over time and eventually leads to electrical failure of the connectors.

On the other hand, fretting due to vibration, dust, the passage of current, temperature, etc., also contributes and accelerates the degradation process of electrical contacts. Swingler et al. [22] reported that in vehicles, the connectors situated close to the exhaust system can possess high-temperature regimes such as 85, 105, 125, and 155 °C. Additionally, Jedrzejczyk [23] experimented on CuSn/Ni/Sn contact material system (Figure 2); he reported that during the process of degradation, electrical resistance depends on the nonoxidized area of contact at the interface. i.e., when all the areas of contact are fully covered by oxidized films, then the electrical resistance value increases above a normal threshold value.

## 2. Analytical Models for Contact Resistance

Numerous studies on contact resistance and electrical contacts have made mention of Greenwood’s and Holm’s analytical model for contact resistance calculation [6,7,8,9,10,11,12,13,14,15,16,17,18]. The research reported in this paper used Holm’s and Greenwood’s models to calculate the contact resistance.

### 2.1. Holm’s Analytical Model for Contact Resistance

In Holm’s analytical model for the calculation of contact resistance [4], two cylinders (C_1_ and C_2_) in contact were considered (Figure 3).

Though both cylinders were assumed to be clean and free of any impurity, they come in contact through a smaller portion of the apparent area (*Aa*). This smaller portion is the actual point of contact (*Ac*). Constriction resistance occurs when the current is restricted from its normal flow to pass through (*Ac*). The voltage between both cylinders can be measured as the current flows through and subsequently the resistance between both surfaces. Holm’s analytical model calculates the total constriction resistance of one circular spot between two electrically conducting cylinders. It has the expression:*R_c_* = ρ∙(2*a*)^−1^,(1)
where ρ is the resistivity of the conductor and *a* is the radius of the constriction.

In situations where there is a difference in material properties (when the bodies in contact have different resistivities), Equation (1) becomes:*R_c_* = (ρ_1_ + ρ_2_)∙(4*a*)^−1^(2)

The assumptions in Holm’s analytical model are:No presence of oxide films or impurities at the interface of the metallic cylinders;There is no axial deviation in the direction of current flow;The metallic cylinders in contact have infinite dimensions to the current flow.

### 2.2. Greenwood’s Analytical Model for Contact Resistance

In 1966, J.A. Greenwood published a paper [5] in which he made a detailed interpretation of Holm’s analytical model. He considered a set of circular spots (multiple a-spots) within a single cluster located at the interface between two electrodes (Figure 4). The metallic electrodes come in contact through the circular spots with no interface film between them. He obtained the formula for calculating the constriction resistance based on multiple spots within a single cluster (3) by treating the current flow between the metallic electrodes similar to that of an electrostatic charge distribution problem.
*R_cG_* = ρ∙(2∑*a_i_*)^−1^ + ρ∙π^−1^ (∑*_i_*_≠*j*_ (*a_i_a_j_*)∙(*s_ij_*)^−1^)/(∑*a_i_*)^2^,(3)
where ρ is the resistivity, s*_ij_* is the distance between the centers of the *i*th and *j*th spot, *a_i_* and *a_j_* is the radius of the *i*th and *j*th spot. The first term ρ∙(2∑*a_i_*)^−1^ is the resistance of all the spots in parallel, the second term ρ∙π^−1^ (∑*_i_*_≠*j*_ (*a_i_a_j_*)∙(*s_ij_*)^−1^)/(∑*a_i_*)^2^ is the resistance due to the interaction between all the spots.

In cases where all the contact spots have the same size, Greenwood further approximated (3) and derived:*R_cG_*_1_ = ρ∙(2∑*a_i_*)^−1^ + ρ∙(π*n*^2^)^−1^ ∑*_i_*_≠*j*_ (*s_ij_*)^−1^(4)

Furthermore, the hypothesis used in Holm’s model also holds in Greenwood’s analytical model.

## 3. Numerical Model

As shown above, the calculation of contact resistance in the case of a discontinuous interface (spot contact) can be done analytically. However, extensive multiphysics studies in which the electrical problem is coupled with thermal and/or mechanical (contact pressure) problems can only be solved numerically. Thus, in this paper, a numerical model is proposed that allows the calculation of the contact spot resistance, but can be further improved and completed later for the study of the thermal regime or the consideration of other parameters that can influence the value of the contact resistance. Therefore, to validate the proposed numerical model, a simple geometry very similar to that treated by Greenwood and consisting of two copper disks communicating through multiple contact spots (Figure 5 and Figure 6) was considered. At the same time, such a model is indicative of the contact between two metal electrodes through a set of circular spots. To attain the simplification of the model, we considered circular spots that are equal in terms of area and relatively evenly distributed across the contact interface. At the terminals, a low direct current (200 mA) of density ***J*** is injected and the electric field distribution is calculated using the finite element method (FEM) in COMSOL Multiphysics software 5.6.

### 3.1. Geometrical Model

The two copper disks model investigated come in contact through multiple spots (28 identical circular spots). Both copper disks have a radius *r* = 5 mm and a thickness *h* = 1 mm, while the homogenous contact spots have a radius *a* = 0.2 mm. Apart from the contact spots, the apparent contact area has an insulating layer of polyethylene 30 µm thick; it restricts the current to flow only through the multiple contact spots (Figure 6).

### 3.2. Mathematical Model

In the COMSOL Multiphysics software, the numerical analysis of the investigated problem was calculated in the stationary electrokinetic regime. It consists of an electromagnetic problem that calculates the distribution of a constant electric current of density ***J*** flowing through the multiple contact spots. The essential equations dictating the problem are the electromagnetic induction law (5), the electric conduction law (6), the electric charge conservation law (7), and the electric field strength, which was evaluated as a function of the electric potential *V* (Equation (8)).
rot ***E*** = 0,(5)
***J*** = σ∙***E***,(6)
div ***J*** = 0,(7)
***E*** = −grad*V,*(8)
where ***E*** is the electric field (*V*/m) and σ = 1/ρ is the electric conductivity (S/m).

#### Boundary Conditions

The boundary conditions applied in this study are:Continuity (9): this signifies that the normal components of the injected current flowing through the copper disks are continuous and conserved across the interior boundaries of both disks: ***n***∙(***J***_1_ − ***J***_2_) = 0(9)Insulation (10): this specifies that no current flows across the boundary. It applies to all surfaces except the contact areas: ***n***∙***J*** = 0(10)

The discretization of the computational domain shown in Figure 7 has a total of 2,938,081 domain elements.

## 4. Results

For an injected current of 200 mA, Figure 8 shows the current lines as they become constricted to flow through the multiple contact spots from one medium of the copper disk to the other.

Figure 9 shows the voltage computed for an injected current of 200 mA, the resistivity of the copper disks (ρ_Cu_ = 1.72∙10^−8^ Ω∙m), the resistivity of the contact spots (ρ_Cs_ = 1.72·10^−2^ Ω·m), and the resistivity of the polyethylene (ρ_PE_ = 10^+17^ Ω∙m). The result shows that for a contact spot radius of 0.1 mm, the maximum voltage is 0.12 V.

After conducting a parametric study on the radius of the contact spots from *a* = 0.1 mm to *a* = 0.5 mm, Figure 10 shows the voltage drop computed for each contact spot radius. As expected, the voltage decrease as the contact spot radius increases. The results show that when the contact spot radius is 0.1 mm, the voltage is 0.12 V; for a radius of 0.3 mm, the voltage is 12.8 mV; and when the radius is 0.5 mm, the voltage becomes 4.6 mV.

Figure 11 shows the contact resistance evaluated in the case where for resistivity of the copper disks is the same as that of the contact spots ρ_Cu_ = ρ_Cs_ = 1.72·10^−8^ Ω·m. The result shows that the contact resistance decreases from 3.25·10^−6^ to 4.34·10^−7^ Ω (numerical), 3.07·10^−6^ to 1.02·10^−6^ Ω (Holms), and 4.76·10^−6^ to 2.71·10^−6^ Ω (Greenwood) when the contact spot radius increases from 0.1 to 0.3 mm. When the contact spot radius increases to 0.5 mm, the contact resistance decreases to 3.47·10^−7^ Ω (numerical), 6.14·10^−7^ Ω (Holm), and, respectively, 2.3·10^−6^ Ω (Greenwood).

Figure 12 shows the contact resistance calculated for resistivity of the copper disks ρ_Cu_ = 1.72·10^−8^ Ω·m, with the resistivity of the contact spots ρ_Cs_ = 1.72·10^−6^ Ω·m (a) and ρ_Cs_ = 1.72·10^−4^ Ω·m (b). The result from Figure 12a shows that the contact resistance decreases from 6.03·10^−5^ to 7.03·10^−6^ Ω (numerical), 3.07·10^−4^ to 1.02·10^−4^ Ω (Holms), and 4.76·10^−4^ to 2.71·10^−4^ Ω (Greenwood) when the contact spot radius increases from 0.1 to 0.3 mm. When the contact spot radius increases to 0.5 mm, the contact resistance decreases to 2.58·10^−6^ Ω (numerical), 6.14·10^−5^ Ω (Holm), and 2.30·10^−4^ Ω (Greenwood).

In Figure 12b, when the contact spot increases from 0.1 to 0.3 mm, the contact resistance decreases from 6·10^−3^ to 6.4·10^−4^ Ω (numerical), 3.07·10^−2^ to 1.02·10^−2^ Ω (Holm), and 4.76·10^−2^ to 2.71·10^−2^ Ω (Greenwood). At a contact spot radius of 0.5 mm, the contact resistance becomes 2.30·10^−4^ Ω (numerical), 6.14·10^−3^ Ω (Holm), and, respectively, 2.30·10^−2^ Ω (Greenwood).

Figure 13 shows the contact resistance obtained for resistivity of the copper disks ρ_Cu_ = 1.72·10^−8^ Ω·m with a resistivity of the contact spots ρ_Cs_ = 1.72·10^+2^ Ω·m (a) and ρ_Cs_ = 1.72·10^+4^ Ω·m (b). In Figure 13a, the result shows that for an increase in contact radius from 0.1 to 0.3 mm, there is a decrease in the value of the contact resistance from 5.25·10^+3^ to 6.39·10^+2^ Ω (numerical), 3.07·10^+4^ to 1.02·10^+4^ Ω (Holms), and 4.76·10^+4^ to 2.71·10^+4^ Ω (Greenwood). For a contact radius of 0.5 mm, the contact resistance becomes 2.30·10^+2^ Ω (numerical), 6.14·10^+3^ Ω (Holm), and, respectively, 2.30·10^+4^ Ω (Greenwood).

The result in Figure 13b indicates that when the contact radius increases from 0.1 to 0.3 mm, the contact resistance value decreases from 5.75·10^+5^ to 6.4·10^+4^ Ω (numerical), 3.07·10^+6^ to 1.02·10^+6^ Ω (Holms), and 4.76·10^+6^ to 2.71·10^+6^ Ω (Greenwood). For a contact radius of 0.5 mm, the contact resistance becomes 2.3·10^+4^ Ω (numerical), 6.14·10^+5^ Ω (Holm), and, respectively, 2.30·10^+6^ Ω (Greenwood).

Figure 14 presents the contact resistance values obtained only by numerical analysis in a case where the resistivity of the copper disks remains ρ_Cu_ = 1.72·10^−8^ Ω·m and the 28 contact spots all have resistivity of ρ_Cs_ = 1.72·10^−6^ Ω·m (■), ρ_Cs_ = 1.72·10^−4^ Ω·m (●), ρ_Cs_ = 1.72·10^−2^ Ω·m (▲), and when the 28 contact spots have different resistivities in the range from 1.72·10^−6^ to 1.72·10^+6^ Ω·m (▼). The resistivity was imposed evenly, i.e., (4 contact spots ρ_Cs1_ = 1.72·10^−^^6^ Ω·m, 4 contact spots ρ_Cs2_ = 1.72·10^−4^ Ω·m, 4 contact spots ρ_Cs3_ = 1.72·10^−2^ Ω·m, 4 contact spots ρ_Cs4_ = 1.72 Ω·m, 4 contact spots ρ_Cs5_ = 1.72·10^+2^ Ω·m, 4 contact spots ρ_Cs6_ = 1.72·10^+4^ Ω·m, and, respectively, 4 contact spots ρ_Cs7_ = 1.72·10^+6^ Ω·m). The results show that the contact resistance value varies between 0.6 Ω (ρ_Cs_ = 1.72·10^−2^ Ω·m at *a* = 0.1 mm) and 2.58·10^−6^ Ω (ρ_Cs_ = 1.72·10^−2^ Ω·m at *a* = 0.5 mm).

## 5. Discussion

As seen in Figure 11, Figure 12 and Figure 13, it is clear that the contact resistance decreases as the contact spot radius increases in all cases; a similar outcome was reported by Fukuyama et al. [18]. He experimented on a configuration consisting of samples with SiO_2_ insulator layers that have gold (Au) contact spots distributed on the circumference of the electrodes. The numerical and analytical results obtained in this study are consistent with the results presented in [18] in terms of the variation of the contact resistance as a function of the contact spots area. However, comparing the numerical calculation of contact resistance to that of the analytical calculation results (Equations (1) and (4)), it is clear that the numerical results are below the two analytical results. Concerning the data presented in [18], it can be noted that the experimental results were between the two analytical calculations (Holm and Greenwood). Additionally, from the point of view of the relationship between contact resistance and the area of contact, similar variation results were reported by Shujuan et al. [13], Ren et al. [14], Lim et al. [15], and Su et al. [24].

In Figure 14, the contact spots with a high resistivity showed a high value of contact resistance and vice versa, as expected. In practice, this is typical of aged electrical contacts where on the one hand, the parts at the interface with a large concentration of contaminants possess high resistivity to the flow of current and thus, have high contact resistance. On the other hand, the parts with fewer/or no contaminants have low contact resistance. Shibata et al. [25] performed a detailed analysis of the contact distribution of fretting corrosion on samples with low phosphorus–bronze alloy covered with tin plating. He reported that the contact resistance distribution results of fretting corrosion wear as observed by scanning electron microscopy (SEM) and energy dispersive X-ray (EDX) analysis indicated that the parts with few oxide deposit layers showed low contact resistance, while those with a large oxide deposit layer showed high contact resistance. It must be noted that both Holm and Greenwood have some drawbacks. In reality, the actual connector consists of multiple small contact areas (a large number of a-spots), but Holm assumed these a-spots to be equivalent to one solid circular contact. On the other hand, Greenwood’s model is suitable for multiple spots within a cluster, and it depends on the distances between the set of the circular spots within that cluster. Though this is an upgrade from Holm’s model, it will be difficult to calculate the constriction resistance when the contact consists of more than one cluster [26].

## 6. Conclusions

This paper presents a study on contact resistance of a model of two metallic disks made of copper that communicate via 28 homogenous circular spots. In the first aspect of the study, the contact resistance calculation was done using Holm’s and Greenwood’s analytical model, and in the second aspect, using a numerical model in COMSOL Multiphysics. The study was carried out keeping the number of contact spots and the electrical resistivity of the copper disks constant and changing the radius and resistivity of the contact spots as a possible aging consequence, between 0.1 and 0.5 mm and 1.72·10^−6^ to 1.72·10^+4^ Ω·m, respectively. As expected, the resulting values of contact resistance in all cases indicate that the contact resistance decreases as the contact spot radius increases; it validates that the contact resistance decreases as the effective contact area increases. This means that the interface contains few weaknesses and the contact pressure between the two conductors takes appropriate values. Additionally, the results show that the value of contact resistance obtained for both analytical models is higher with at least an order of magnitude than that of the value obtained by numerical analysis in all cases.

Furthermore, without considering the model used or the contact spot radius, as the resistivity of the contact spot increases in all cases (Holm, Greenwood, and numerical), the contact resistance value obtained increases by two orders of magnitude. In the future, detailed work on the electrothermal coupling of the model will be done to help understand the thermal regime and its effect on contacts.

## Figures and Tables

**Figure 1 materials-15-02056-f001:**
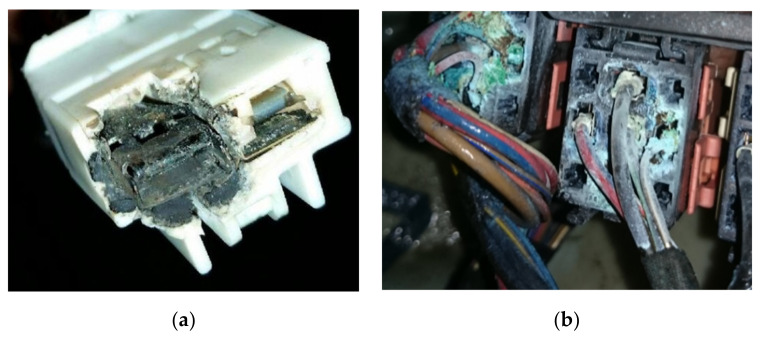
Examples of corroded connectors: (**a**) corroded Ac blower wire harness connector in an Acura TL [20]; (**b**) corroded wiper control relays in a BMW 7 series [21].

**Figure 2 materials-15-02056-f002:**
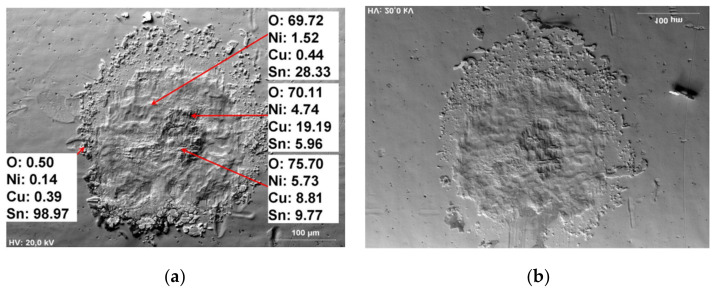
Fretting scars as seen in an SEM micrograph obtained after 2600 fretting cycles in a CuSn/Ni/Sn contact material system: (**a**) lower sample (**b**) upper sample [23].

**Figure 3 materials-15-02056-f003:**
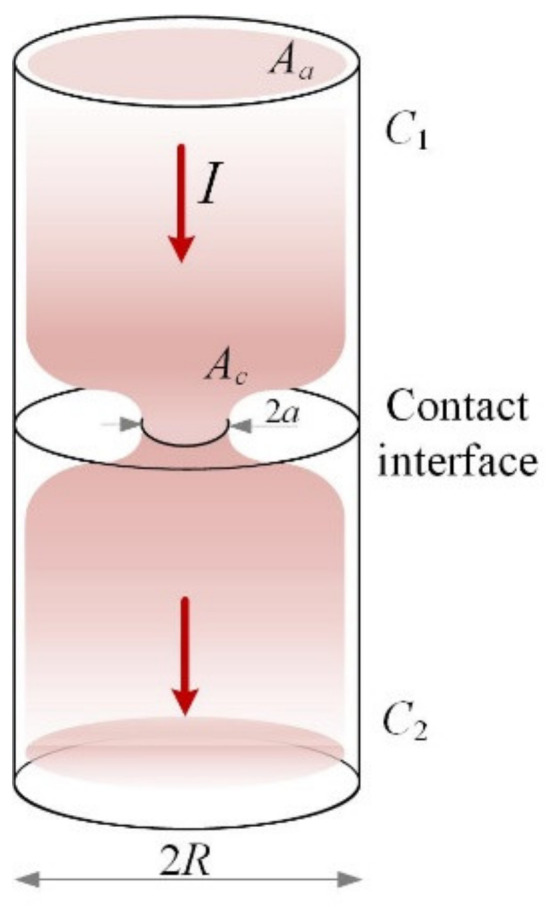
Holm contact resistance model: cylinders in contact showing the apparent area of contact (*Aa*), the real area of contact (*Ac*), and the radius *R* with a circular constriction of the current lines of radius *a*.

**Figure 4 materials-15-02056-f004:**
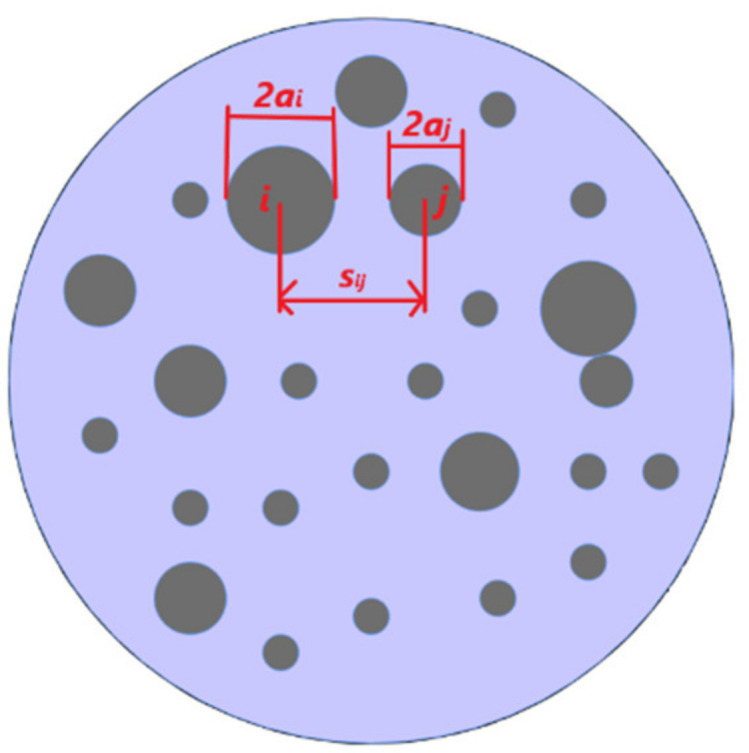
Greenwood’s contact resistance model showing multiple spots (*a_i_*—radius of the *i*th spot; *a_j_*—radius of the *j*th spot; s*_ij_* the distance between the *i*th and *j*th spot).

**Figure 5 materials-15-02056-f005:**
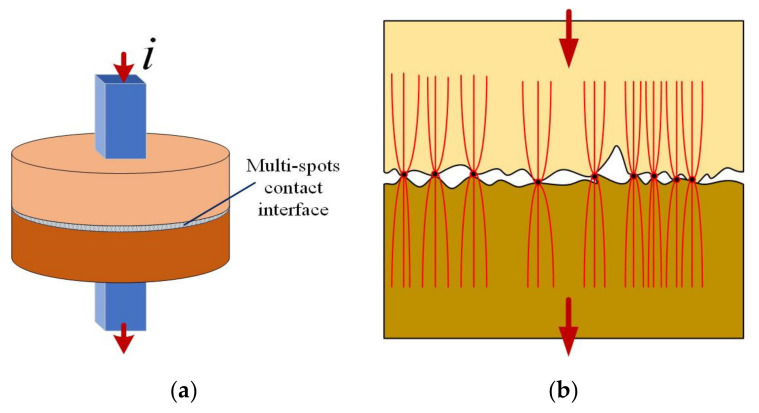
Schematic view of the contact configuration: (**a**) copper disks in contact; (**b**) constriction of the current lines flowing through the multiple contact spots.

**Figure 6 materials-15-02056-f006:**
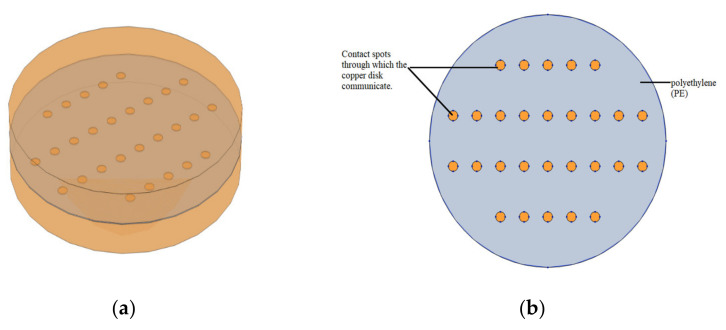
Geometrical model of the problem investigated: (**a**) two copper disks in contact; (**b**) contact spots and the insulating layer of polyethylene (PE) at the interface.

**Figure 7 materials-15-02056-f007:**
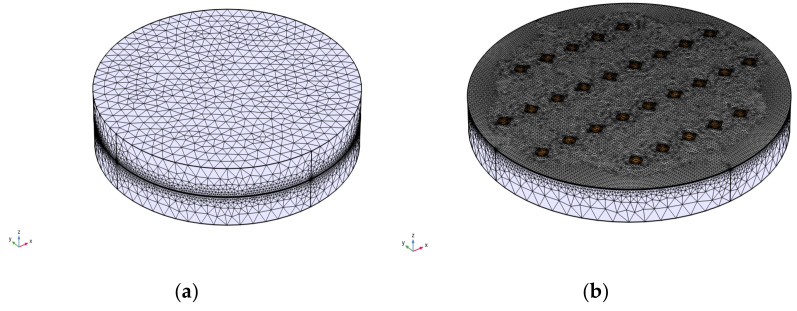
Discretization of the computational domain: (**a**) two copper disks in contact; (**b**) contact spots and the insulating layer of polyethylene (PE) at the interface.

**Figure 8 materials-15-02056-f008:**
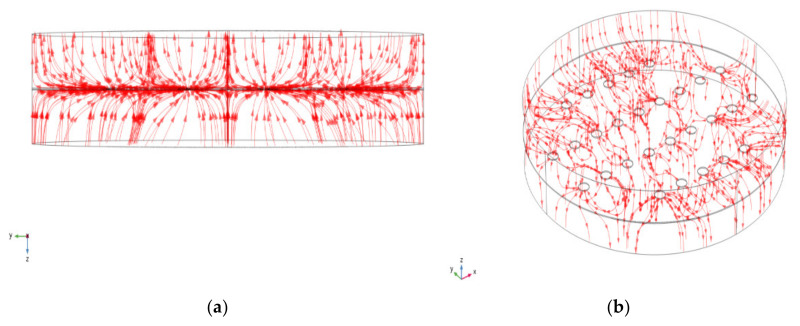
The current line constricted at the contact spots: (**a**) two-dimensional view and (**b**) three-dimensional view.

**Figure 9 materials-15-02056-f009:**
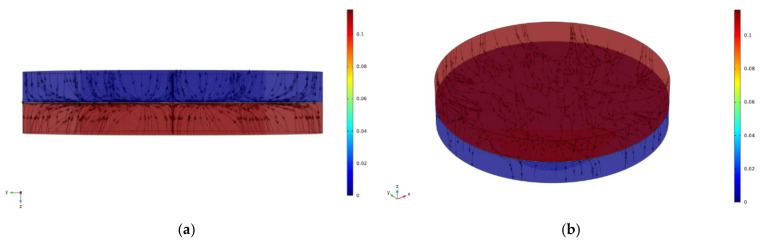
Computed voltage for contact spot radius of 0.1 mm: (**a**) two-dimensional view and (**b**) three-dimensional view.

**Figure 10 materials-15-02056-f010:**
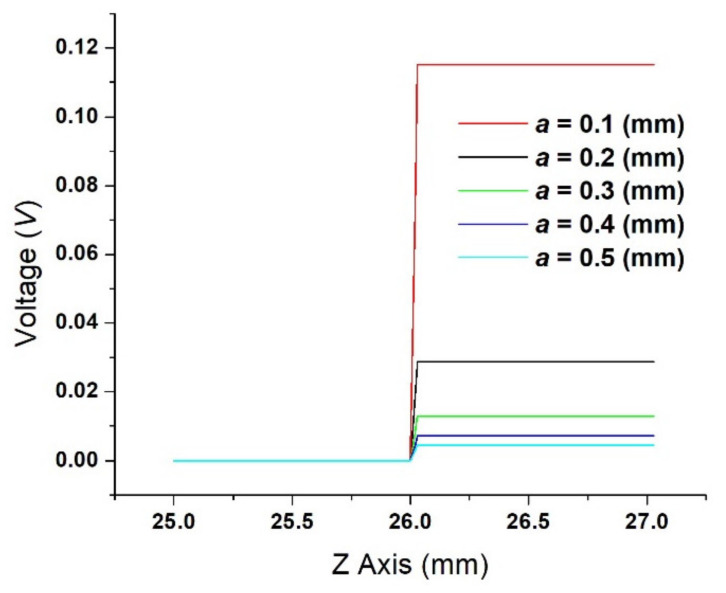
Voltage-drop for different values of contact spot radius *a* = (0.1 mm/0.5 mm).

**Figure 11 materials-15-02056-f011:**
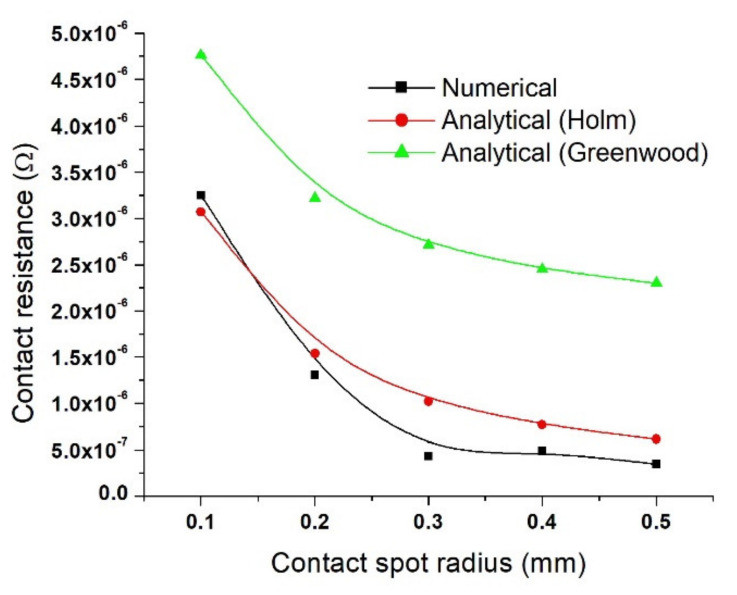
Contact resistance showing values for Holm and Greenwood (analytical) and numerical analysis calculated for (ρ_Cu_ = ρ_Cs_ = 1.72·10^−8^ Ω·m).

**Figure 12 materials-15-02056-f012:**
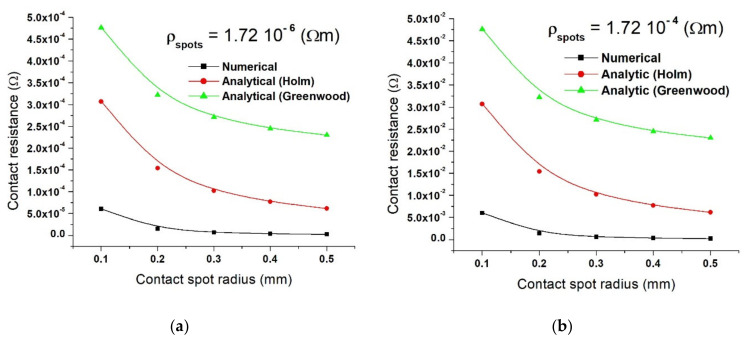
Contact resistance showing values for Holm and Greenwood (analytical) and numerical analysis calculated for ρ_Cu_ = 1.72·10^−8^ Ω·m with (**a**) ρ_Cs_ = 1.72·10^−6^ Ω·m and (**b**) ρ_Cs_ = 1.72·10^−4^ Ω·m.

**Figure 13 materials-15-02056-f013:**
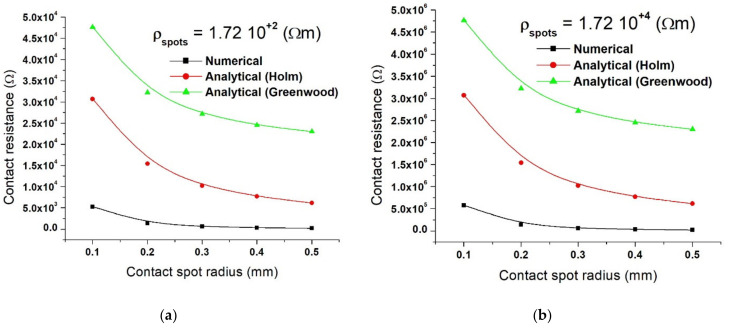
Contact resistance showing values for Holm and Greenwood models and numerical analysis calculated for ρ_Cu_ = 1.72·10^−8^ Ω·m, ρ_Cs_ = 1.72·10^+2^ Ω·m (**a**), and ρ_Cs_ = 1.72·10^+4^ Ω·m (**b**).

**Figure 14 materials-15-02056-f014:**
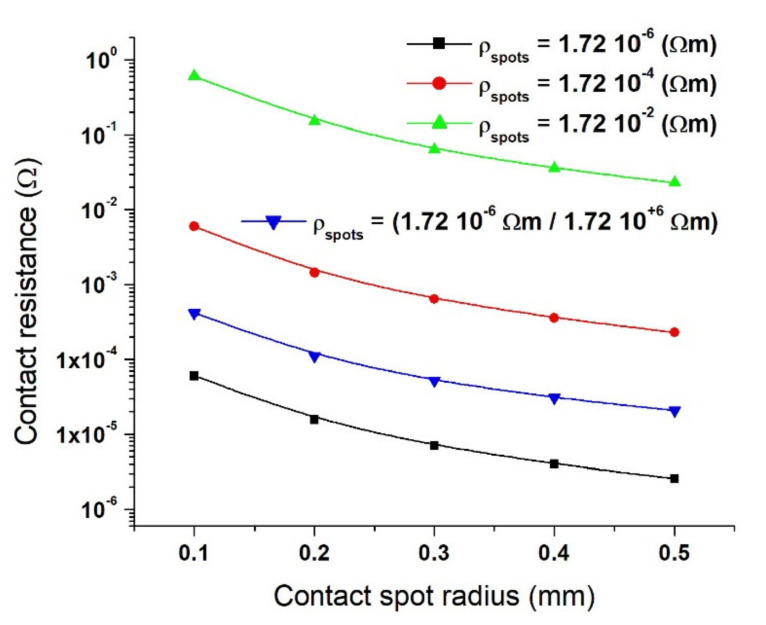
Contact resistance showing values for numerical model calculated for ρ_Cu_ = 1.72·10^−8^ Ω·m with ρ_Cs_ = 1.72·10^−6^ Ω·m; ρ_Cs_ = 1.72·10^−4^ Ω·m; ρ_Cs_ = 1.72·10^−2^ Ω·m, and, respectively, ρ_Cs_ = (1.72·10^−6^ Ω·m/1.72·10^+6^ Ω·m).

## Data Availability

Not applicable.

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
