# Peer review of "Computation of the Electrical Resistance of a Low Current Multi-Spot Contact"

_materials, 2022, doi:10.3390/ma15062056_

Round 1

Reviewer 1 Report

The paper introduces a comparison of calculations of contact resistance. The authors compare the results of two analytical formulas with a numerical calculation. The results are interesting, but the presentation of the work requires major improvement in three points:
-First: the literature on contact resistance is very big. Therefore a deeper literature review is necessary. Consequently, the section 'Introduction' requires improvement. Please review the literature on contact resistance and explain why the Holms (Maxwell) and Greenwood equations were chosen for analytical reference.
-Second: Please explain why the model of Fig. 7 is used for numerical modelling? Introduce the considerations behind it.
-Third: Real discussion of the results is missing from the paper. The section 'Results and Discussion' is only the presentation of the results. However, the presentation of the results is correct. Please discuss the results in the mirror of the existing literature.

Reviewer 2 Report

This manuscript is entitled Computation of the Effect of Aging Process on the Electrical Contact Resistance of a Low Current Multi-Spot Contact. These data are interesting, but some points still need to correct before publication.
1. The English Language must be polished throughout the text (including Title).
2. Remove black colour background from Figure 5.
3. Some places fig and figure, please check throughout the manuscript
4. Please check all units in order to be similar
5. It is suggested to add previous studies results that can help to demonstrate how current
systems good are.

Reviewer 3 Report

I think the manuscript is suitable for publication in the Materials.

It will strength the paper if the authors can explain the differences between several models and given the application scenarios and drawbacks of the different models.

Typo: page 9 line 207, it should be fig. 13a not fig. 11a 

Round 2

Reviewer 1 Report

The paper has been significantly improved. I have no further questions or comments on the manuscript.

Reviewer 2 Report

Well Done.